# Development of PCR-Based Race-Specific Markers for Differentiation of Indian *Fusarium oxysporum* f. sp. *cubense*, the Causal Agent of Fusarium Wilt in Banana

**DOI:** 10.3390/jof8010053

**Published:** 2022-01-05

**Authors:** Raman Thangavelu, Esack Edwinraj, Muthukathan Gopi, Periyasamy Pushpakanth, Kotteswaran Sharmila, Manivasakan Prabaharan, Murugan Loganathan, Subbaraya Uma

**Affiliations:** 1ICAR—National Research Center for Banana, Plant Pathology Division, Tiruchirappalli 620102, Tamil Nadu, India; edwinrht@gmail.com (E.E.); gopimusa@gmail.com (M.G.); Lalitha_pushpakanth@yahoo.com (P.P.); sharmilakotteswaran@gmail.com (K.S.); praba.nrcb@gmail.com (M.P.); logumuruga@gmail.com (M.L.); umabinit@yahoo.co.in (S.U.); 2Research and Development Division, MIRO Forestry SL Ltd., Mile 91, Tonkolili District, Sierra Leone; 3Department of Bioinformatics, Tamil Nadu Agriculture University, Lawley Road, Coimbatore 641003, Tamil Nadu, India

**Keywords:** banana, Cavendish, *Foc*, Fusarium wilt, markers, Polymerase Chain Reaction

## Abstract

Fusarium wilt caused by *Fusarium oxysporum* f. sp. *cubense* (*Foc*), is the most lethal soil-borne fungal pathogen infecting bananas. *Foc* race 1 (R1) and 4 (R4) are the two most predominant races affecting the economically important Cavendish group of bananas in India. A total of seven vegetative compatibility groups (VCGs) from three pathogenic races were isolated during our field survey and were found to be highly virulent towards cv. Grande Naine. According to comparative genome analyses, these Indian *Foc* VCGs were diverse in genomic organization and effector gene profiles. As a result, false-positive results were obtained with currently available molecular markers. In this context, the study has been initiated to develop PCR-based molecular markers for the unambiguous identification of Indian *Foc* R1 and R4 isolates. Whole-genome sequences of *Foc* R1 (GCA_011316005.3), *Foc* TR4 (GCA_014282265.3), and *Foc* STR4 (GCA_016802205.1), as well as the reference genomes of *Foc* (ASM799451v1) and *F. oxysporum* f. sp. *lycopersici* (*Fol*; ASM14995v2), were aligned to identify unique variable regions among the *Foc* races. Using putative chromosome and predicted gene comparison, race-specific unique *Foc* virulence genes were identified. The putative lineage-specific identified genes encoding products secreted in xylem (SIX) that may be necessary for disease development in the banana. An in silico analysis was performed and primers were designed from a region where sequences were dissimilar with other races to develop a specific marker for *Foc* R1, R4, TR4, and STR4. These race-specific markers allowed target amplification in the characterized highly virulent *Foc* isolates, and did not show any cross-amplification to any other *Foc* races, VCGs or banana pathogens, *Fusarium* species, and non-pathogenic *Fusarium oxysporum* isolates. The study demonstrated that the molecular markers developed for all the three *Foc* races of India could detect the pathogen in planta and up to 0.025 pg µL^−1^ DNA levels. Thus, the markers developed in this study are novel and could potentially be useful for the accurate diagnosis and detection of the Indian *Foc* races which are important for the effective management of the disease.

## 1. Introduction

Banana and plantain (*Musa* spp.) are the most important fruit crop in the world and approximately 148 million tones are produced annually from 135 countries [1]. It serves as a staple food for more than 400 million people worldwide and a source of income to many developing countries. Although several hundreds of varieties are grown in different parts of the world, the Cavendish group of bananas contributes to 47% of global production, with an export of 22.3 million tones which worth US$ 13.8 billion per annum. However, Fusarium wilt is caused by a soil-borne fungal disease *Fusarium oxysporum* f. sp. *cubense* (*Foc*) and is becoming a major limiting factor of banana production worldwide. The chlamydospores produced by the *Foc* allow the fungus to thrive in the soil in the absence of a host, and susceptible varieties were unable to be cultivated economically for decades [2,3]. The symptoms produced by this Fusarium wilt pathogen include chlorosis in leaves, pseudostem splitting with internal vascular discoloration, and eventually death of the entire plants.

Based on the pathogenicity towards susceptible banana cultivars, *Foc* is classified into three races: race 1 (R1), race 2 (R2), and race 4 (R4). The *Foc* R1 attack mainly the variety Gros Michel (AAA), Silk (AAB), Pome (AAB), and Pisang Awak (ABB), *Foc* R2 attacks cooking type (ABB) of bananas and R4 (Tropical Race 4 -TR4; and sub-Tropical Race 4-STR4) attacks Cavendish group and also the R1 and R2 susceptible varieties of bananas. Among the two types of *Foc* R4, the *Foc* STR4 attacks Cavendish in the subtropics, and the *Foc* TR4 attacks bananas grown in both tropics and subtropics. Among all these races, the *Foc* TR4 (vegetative compatibility group (VCG) 01213/16) is considered as the most destructive because of its much broader host range and high virulence compared to other strains of *Foc* races. At present this lethal strain has spread to 21 different countries, namely Taiwan, Malaysia, Indonesia, China, Australia, Philippines, Oman, Jordan, Mozambique, Lebanon, Laos, Vietnam, Myanmar, Israel, India, Colombia, Thailand, Turkey, Mayotte, Peru, and the United Kingdom [4]. A report from FAO says that by 2028, globally, the *Foc* TR4 damage to bananas would lead to a loss of 160,000 ha of banana cultivation area, 2.8 million tonnes of banana production, a loss of direct employment for approximately 240,000 banana workers, and a 9.2% rise in the global reference price of bananas [5].

In India, 30.86 million tonnes of bananas are produced annually, which contributes 21% to global production [6]. About 20 different varieties of bananas are commercially grown in an area of approximately 860,000 ha [4]. These include Cavendish (AAA), Rasthali (Silk-AAB), Ney Poovan (AB), Karpuravalli (Pisang Awak-ABB), Monthan (Cooking bananas-ABB), Hill banana (Pome-AAB), Poovan (Mysore-AAB), Red banana (AAA), etc. Among these, the Cavendish group of bananas alone occupies 57% of the cultivated area and most of the export (90%) is from Cavendish banana. Unfortunately, Fusarium wilt is becoming a major production constraint in almost all the banana-growing regions of India, as in the present world situation. A recent survey revealed the incidence in Cavendish banana range from 1% to 45% and sometimes >70%, resulting in the abandonment of the banana fields, switching over to non-remunerative crops like maize (only ₹25,000 as against ₹2–3 lakhs from the banana), and migration of banana workers to distant states of India for the employment [7]. The *Foc* samples collected from Cavendish banana grown in different states of India indicated the presence VCGs such as 0124, 0125, 01220 of *Foc* R1, 0120 of *Foc* STR4, and 01213/16 of *Foc* TR4 which all are found to attack most of the commercial varieties of banana including Cavendish bananas. Interestingly the VCGs, 0125 and 01220 attacking Cavendish banana were identified in the state of Bihar and Uttar Pradesh, where the Fusarium wilt TR4 was also reported [8]. At present, these Cavendish infecting strains of *Foc* are widely distributed across India viz., Tamil Nadu, Kerala, Gujarat, Maharashtra, Madhya Pradesh, West Bengal, Uttar Pradesh, and Bihar [9], while VCG0120 (STR4) was reported from Gujarat and Madhya Pradesh, where the Cavendish is grown as a monoculture (Figure 1).

To date, only a few effective control strategies have been devised to protect the bananas from Fusarium wilt; nevertheless, their effectiveness remains minimal [10]. In this context, the best strategy to manage this disease is to use resistant genotypes [11], however, these are not always preferred by the farmers/consumers. Under these circumstances, the only way to prevent the spread of this disease is to implement adequate phytosanitary measures and early detection of the pathogen in the non-symptomatic infected plants, contaminated water, and soil [12]. The traditional method of identifying *Foc* pathogen and its races is based on symptomatology, culture morphology, and microscopic examination of the mycelia and spore characters which is time-consuming, destructive, and requires expertise [13,14,15]. The widely used VCG method for identifying *Foc* strains also requires 15–20 days, but it gives no indication of genetic relatedness between different VCGs and is ineffective for some *Foc* isolates which in this case are classified as unknown VCGs. Moreover, there are no differentials available to distinguish *Foc* races based on pathogenicity on different varieties of banana as some *Foc* strains isolated from Gros Michel (a differential cultivar of *Foc* R1) may cause disease in cooking type bananas (differential cultivar to *Foc* R2) and *Foc* strains isolated from cooking type bananas may cause disease in *Foc* R1 susceptible cultivars (Rasthali AAB) [8,16,17]. A recent study found that the *Foc* VCGs 0125 and 01220 belonging to *Foc* R1 could cause disease in Cavendish banana which is considered as *Foc* R4 differential (8). Therefore, to address the aforesaid challenges in pathogen identification, molecular markers that are more precise and reproducible are desperately needed.

Several molecular techniques have been developed so far which are paramount for the initial detection, identification and race determination of *Foc* isolates from various geographical regions. Numerous primers targeting various genes (e.g., ef1-α, IGS, SIX, etc.) and Polymerase Chain Reaction (PCR) techniques, such as conventional PCR, quantitative PCR (qPCR), real-time fluorescent loop-mediated isothermal amplification (LAMP), and diversity arrays technology sequencing (DArTseq) have been established [18,19,20,21]. Some techniques and primers are more effective than others in recognizing *Foc* races or VCGs. PCR-based techniques such as RAPD, SCAR, AFLPs, DAF, and SNP markers have been employed by García-Bastidas et al. [22]; Ploetz, [23]; Ordóñez et al. [24]; Chittarath et al. [25]; Hung et al. [26]; Zheng et al. [27]; Thangavelu et al. [28] to recognize different races of *Foc*. Similarly, the qPCR method has been developed for race typing *Foc* isolates which enables absolute quantification of pathogen DNA at low concentrations, and often samples from the plant, water, and soil are used to diagnose *Foc* R4 [29] and TR4 [20]. LAMP is a portable and easy-to-use quantitative detection method developed for the detection of *Foc* R4 [18] and TR4 [30]. DArTseq, a genotyping technology, was used to obtain DNA markers unique to *Foc* TR4 isolates [30]. The *Foc*4 0422F2/R2/P2 probe-based test, described as a *Foc* R4 detection tool by Yang et al. [31], was found to be specific and sensitive in targeting both VCGs 01213/16 and 0121. Li et al. [32] identified a potential virulence gene for diagnosis *Foc* TR4 through T-DNA insertional mutagenesis and developed a marker based on thermal asymmetric interlaced-PCR (TAIL-PCR) analysis.

These aforesaid molecular diagnostic methods were generally based on genes and core genomic regions and are unlikely to be closely associated with pathogenic characteristics of each race. The *Foc* TR4 diagnostic method developed based on the intergenic spacer region (IGS) of the nuclear ribosomal gene cluster of *Foc* is being widely used across the world [19]. However, this detection assay was not designed to detect VCGs which are closely related to TR4 and can also be able to infect Cavendish cultivars [24,33,34]. Besides, we found in a laboratory experiment that this marker also gives a false-positive result with *Foc* R1 isolated from cv. Ney Poovan (AB), Rasthali (Silk AAB) and Pisang Awak (ABB). Earlier, Lin et al. [21] developed molecular markers to detect *Foc* R4 strains by using random primers. Moreover, the homologs of secreted in the xylem (*SIX*) genes which contribute to the virulence of various formae specialis of *Fusarium oxysporum* (*Fo*) are also being exploited for diagnostic purposes and the VCGs of *Foc* vary in SIX protein profile and their respective gene sequence [34,35,36,37,38,39]. As there are strong links between virulence genes and pathogenicity, they have been regarded as excellent markers for host-specific *Foc* strains [32]. In this direction, Carvalhais et al. [40] developed diagnostic assays based on conventional PCR targeting *SIX* genes which accurately detected *Foc* R1, STR4, and TR4 as well as two VCGs (0121 and 0122) causing Fusarium wilt in bananas. But unfortunately, the marker developed by Carvalhais et al. [40] for *Foc* R1 had given a positive result for the *Foc* STR4 (Personnel observation). Therefore, developing unambiguous *Foc* race-specific molecular markers for Indian *Foc* isolates infecting Cavendish bananas is a pressing requirement. For this, we exploited the whole-genome sequences of all the *Foc* races of India and successfully developed molecular markers from the sequences of *SIX* genes (for *Foc* TR4, STR4, and R4) and a hypothetical effector protein associated with virulence (for *Foc* R1). Although just a few *Foc* specific primers are currently available, the simultaneous use of these primers will ensure not only the accuracy, but also false-positive or negative results do not occur [41] in diagnosing and detecting Indian *Foc* isolates which have variations in whole-genome sequences. Therefore, the study aimed to: (1) identify virulence-associated genes with putative roles in Fusarium wilt from whole-genome sequences published by Thangavelu et al. [42]; (2) find a unique variable region that could be used to design markers for different races (*Foc* R1, R4, TR4, and STR4) for specific detection; and (3) determine the specificity and sensitivity of the newly developed primers in identifying highly virulent Indian *Foc* isolates amongst characterized and uncharacterized *Fo* populations collected from various locations of India.

## 2. Materials and Methods

### 2.1. Collection of Foc Races

A total of 61 *Foc* isolates were isolated from different banana-growing states of India (11 isolates from Uttar Pradesh, 5 from Gujarat, 13 from Bihar, 23 from Tamil Nadu, 5 from Kerala, and 4 from Madhya Pradesh; Figure 1 and Table 1). The race typing of *Foc* was confirmed by pathogenicity tests on the respective banana cultivars and also on the cv. Grande Naine. Further, the genomic DNA of 23 reference *Foc* isolates (Appendix A) of different VCGs (obtained from QDPI and F, Brisbane, Australia), five isolates of nonpathogenic *Fo*, one each of *Pseudocercospora eumusae*, *Colletotrichum musae*, *Fusarium solani*, and *Klebsiella variicola* obtained from Department of Plant Pathology, ICAR-National Research Center for Banana, Tiruchirappalli was included in the study (Table 1).

All Indian *Foc* races of different VCGs employed in the present study were isolated from the vascular strand of pseudostem of a symptomatic Grande Naine cv. (Cavendish AAA subgroup). VCG analysis with *nit*-M (Nitrate non-utilizing mutants) testers [43], in planta pathogenicity testing using micropropagated Grande Naine plantlets, and molecular confirmation with race-specific PCR markers (e.g., [19,40,44]) were performed to reconfirm the virulence and races of the isolate before the whole-genome sequencing. Thangavelu et al. [28] furnished a detailed description of the procedures elsewhere.

### 2.2. DNA Extraction and Quantification

Genomic DNA was isolated according to the method described by Dellaporta et al. [45] with slight modifications. Freeze-dried fungal mycelium (1 g) and overnight grown bacterial culture (2.0 mL) were frozen in liquid nitrogen and ground to fine powders using a precooled mortar and pestle. DNA was extracted using 5 mL modified TNE buffer (100 mM Tris-HCl, pH 8.0; 50 mM Na_2_EDTA, pH 8.0; 50 mM NaCl; 8 μM β-mercaptoethanol; 1% SDS and 10 μg mL^−1^ RNase) and incubated at 65 °C for 1 h. A 0.33× volume of 5 M CH_3_CO_2_K was added, mixed, and centrifuged at 10,000 rpm for 5 min. The supernatant was transferred into a new tube and mixed with an equal volume of isopropanol to precipitate crude DNA. The samples were incubated at −20 °C overnight and centrifuged at 4 °C at 10,000 rpm for 10 min. After resuspending the DNA pellet in 300 μL dH_2_O, an equal volume of phenol-chloroform/isoamyl alcohol (25:24:1; *v*/*v*) was added and mixed thoroughly. The upper aqueous layer was transferred to a fresh tube after centrifugation at 10,000 rpm for 5 min at 4 °C, and 0.1 × volume of 5 M potassium acetate and 2.5 × volume of absolute ethanol (70%) were added. The supernatant was decanted following the centrifugation at 10,000 rpm for 10 min at 4 °C, and the DNA pellet was washed with 300 μL of 75% ethanol, left to air dry, and then finally dissolved in 1 × TE buffer (10 mM Tris-HCl, pH 8; 0.1 mM EDTA) for further analysis. Total DNA yield and purity were quantified by measuring OD at 260 nm using the NanoDrop LITE (Thermo Scientific, Waltham, MA, USA).

### 2.3. Sequence Alignment and Primer Design

Genomic sequences of *Foc* isolates (PRJNA552452, PRJNA687504 and accession numbers SAMN12206536, SAMN12206882, and SAMN17147622) along with the reference genomes of *Fol*4287 (ASM14995v2; txid426428) and *Foc* TR4 (ASM799451v1; txid61366) were obtained from the National Center for Biotechnology Information (NCBI). The Interactive Genomics Viewer was used to visualize and align the whole-genome sequences. Genome-wide distribution of DNA polymorphisms was initially analyzed by calculating their frequency of every 100 kb interval of reference genomes at chromosome levels. To identify the potential pathogenicity-related proteins, a Basic Local Alignment Search Tool (BLASTp) search was performed against the Pathogen–Host Interaction database (PHI-base) with a threshold E-value of ≤1 × 10^−5^ [46]. SIX genes in the genome of Indian *Foc* races are defined using BLAST analysis with the assembled genome to SIX genes in the NCBI database [47,48]. Targets for primer design were searched on differential regions of the genome, as these contained previously targeted gene candidates for race differentiation. Regions specific to each race were targeted specifically. Using this approach, different primer sequences were isolated as candidates for differentiation based on their genome. NCBI BLAST was used to determine the likelihood of genetic conservation within the genome based on shared sequences with related species to narrow the possible pool of target loci. In addition, sequences containing coding regions and hypothetical proteins were explicitly targeted specifically for primer design. Following BLAST analysis, primer design properties such as GC content and amplicon size were used to narrow further. Consequently, primer pairs for *Foc* R1, R4, TR4, and STR4 were designed manually and checked for quality and content with the Integrated DNA Technologies PrimerQuest™ Tool. All primers were synthesized by Sigma-Aldrich^®^ (St. Louis, MO, USA) and stored at −20 °C. All primers used in this study are listed in Appendix A.

### 2.4. Primer Design and Polymerase Chain Reaction (PCR) Conditions

To achieve this goal, the whole-genome sequences of three races of *Foc* (R1, TR4, and STR4), as well as two other *F. oxysporum* (*Fol, Foc*) from NCBI were used for in silico analyses (www.ncbi.nlm.nih.gov accessed on 4 January 2020). The genome comparison finds the unique effector genes and variable regions between race/VCGs through *SIX* sequences from different *Foc* races were found to be highly conserved in general. When the homology of different races was compared by using a comprehensive suite of molecular biology and next-generation sequencing analysis tools, our target races, (*Foc* R1 and R4) revealed variations in a certain portion of sequences at different locations on chromosome 14 compared to all other races. The variable sequences facilitated the development of *Foc* R1 and R4 markers.

The primers were designed to target *SIX* gene sequences extracted from the whole-genome of Indian *Foc* races using Primer3 software [47] and were tested for specificity in silico using ‘In silico simulation of molecular biology experiments software’ (www.insilico.ehu.es accessed on 4 January 2020). Only variable regions were chosen over highly conserved regions as they may possess race-specific signature sequences required for identifying *Foc* R1, R4, TR4, and STR4. A total of 46 potential race-specific primer sets were designed to discriminate the *Foc* races in which 32 were for TR4, 1 for R4, 9 for R1, and 4 for STR4. Table 2 shows the specific region and expected amplification for these selected *Foc* primers that were cull out from variable regions of the genome. EmeraldAmp^®^ PCR Master Mix (Takara, Shiga, Japan) was used in Polymerase Chain Reaction (PCR) to amplify the target regions with respective markers. A 20 µL of the PCR reaction mixture containing forward and reverse primers (0.5 µM µL^−1^ each), EmeraldAmp^®^ PCR master mix (9.0 µL), ultra-pure water (8 µL), and 50 ng µL^−1^ DNA was used for PCR amplification. PCR was performed using the following conditions in a thermocycler (Takara, Shiga, Japan): denaturation at 95 °C for 5 min followed by 30 cycles of amplification at 95 °C for 30 s, annealing at 65 °C for 40 s and 72 °C for 45 s, and final elongation at 72 °C for 5 min, but in case of *Foc* TR4, R4 and STR4 specific amplification annealing was carried out at 66 °C for 40 s. Electrophoresis was undertaken using 2% agarose gel at 100 V for 30 min and visualized on the gel documentation system under ultraviolet light (302 nm).

### 2.5. Optimization of the Developed Marker

All the new primer sets were evaluated to determine the optimal annealing temperature, the level of specificity, and the detection limit of genomic DNA. To optimize the annealing temperature, a gradient PCR from 58 to 68 °C was performed and the highest temperature that did not diminish the brightness of the amplicon was selected. To assess specificity, each primer set was tested with a diverse selection of nonpathogenic *Fo* isolates, *F. solani*, *Foc* VCGs of all three races, and other banana pathogens such as *P. eumusae*, *C. musae*, and *K. variicola*. PCR samples were prepared to the same concentration levels as described above, with approximately 25 ng of DNA per reaction. Along with the non-target samples, a positive *Foc* control and negative water control were also included. To evaluate the sensitivity of the newly developed PCR marker, genomic DNA was isolated from various standard *Foc* isolates and standardized to 25 ng µL^−1^, then serially diluted down to 0. 02 pg µL^−1^ of DNA. DNA concentrations were quantified in a Qubit fluorometer using the Quant-iT High-Sensitivity dsDNA Assay Kit (Thermo Fisher Scientific Inc., Waltham, MA, USA) before dilution. To determine the minimum sensitivity, these dilutions were amplified using the specified race-specific primers. PCR results (4 µL per sample) were imaged on a 2% agarose gel and the lowest successful amplification was determined to be the detection limit of the new primer set. Reproducibility of the PCR assay was further confirmed by two different operators, on three different occasions, with the same set of samples. All tests included two technical replicates, and the results demonstrated that the primers met all repeatability and robustness validation criteria.

### 2.6. In Planta Detection of Foc Races

The experiment used three-month-old tissue-cultured plants of cv. Grande Naine and planted in poly grow bags (30 × 10 × 15 cm) filled with a sterilized potting mixture containing red earth, sand, and farmyard cattle manure in equal proportions and were kept under glasshouse conditions for 10-days for initial acclimatization. Fifteen-day-old pure *Foc* R1 (VCG 0125, 0124 and 01220) and *Foc* R4 (TR4 and STR4) cultures multiplied individually in the sand: maize (19:1) medium containing approximately 10^6^ CFU g^−1^ was applied once in the soil around the plants in each plant at a 1 inch depth. The experiment was conducted as a randomized complete block design with three replications and each replication containing five plants. These inoculated plants were maintained in a glasshouse at 28 °C, 80% relative humidity, and 16 h light. After 25 days of inoculation, the rhizome and pseudostem were subjected to systematic examination by split into two halves and disease evaluation was carried out based on the percentage of the discolored area in the corm on a 0–5 scale [49]. Samples from the first half of the rhizome and vascular strands were plated on Komada’s medium for selective isolation of *Foc* and confirmation and the other half was used for genomic DNA extraction for in planta detection of *Foc* races. Total genomic DNA from 1 g of vascular strands and corm was extracted using the DNeasy^®^ plant mini kit (Qiagen Cat no. 69104) according to the manufacturer’s instructions. In planta detection was performed using the respective primers designed as mentioned in Table 2 to amplify *Foc* strains of R1, R4, TR4, and STR4.

## 3. Results

The genome of Indian *Foc* races was assembled at chromosome-scale and the genome size was ranged from 61 to 63 Mb with ~48.5% GC. The genome of Indian *Foc* TR4 (VCG01213/16) is 16.3–26.4% higher in size in terms of nt bp, whereas, *Foc* R1 is 16.8–22.5% and *Foc* STR4 is 23.9% higher than the reference genome. A total of 21,842 intact protein-coding genes were predicted from the consensus gene sets of the *Foc* R1 genome assembly, while, the *Foc* STR4 and TR4 genome assemblies contain 17,118 and 17,745 protein-coding regions, respectively. The results revealed that the *Foc* R1 contained a maximum of 524 tRNAs and 223 rRNAs, TR4 has a maximum of 358 tRNAs and 134 rRNAs, and STR4 has a total of 304 tRNAs and 121 rRNAs in the genome. A total of 8756 (39.5%) out of 22,151 proteins coding genes in *Foc* R1, 5384 (28.4%) out of 18,946 protein-coding genes in *Foc* STR4, and 11,220 (57.1%) out of 19,651 protein-coding genes from *Foc* TR4 were annotated from the Gene Ontology database with a cutoff E-value of 10^−5^. A total of 4866 high-quality SNPs were found within the Indian *Foc* isolates, in which 1502 in *Foc* R1, 1844 in *Foc* STR4 and 1520 in *Foc* TR4, while a total of 139,023 InDels (88,532 insertions and 50,491 deletions) in which 53,602 in *Foc* R1, 33,298 in *Foc* STR4 and 52,123 in *Foc* TR4. The study identified 1738 (*Foc* R1) to 2809 (*Foc* STR4) putative virulence-associated genes in the genome of Indian *Foc* isolates in which a total of 94, 283, and 258 virulence genes are unique to the *Foc* R1, TR4, and STR4 genomes, respectively. The number of effector genes encoded by the genome varies in which the lowest 17 genes were present in *Foc* TR4, a moderate 20 genes in *Foc* R1, and a maximum of 28 genes in *Foc* STR4. Similarly, the number of genes involved in increased virulence was 13, 23, and 27 in *Foc* TR4, STR4, and R1, respectively. These virulence genes were the target for the primer synthesis for the race-specific identification of the Indian *Foc* isolates.

### 3.1. Screening of Race-Specific Markers and Its Specificity to Indian Foc

Out of 61 *Foc* isolates, 27 representative isolates of different races were subjected to the specificity analysis. Out of nine primers tested; the *Foc* R1F/*Foc* R1R primer set targeted to gene XM_018394505.1 encoding a hypothetical protein (Table 2) resulted in 320 bp amplicon only with *Foc* R1 VCGs 0124, 0125, and 01220 (Figure 2A). In the case of *Foc* R4, only one primer set *Foc*R4F/*Foc*R4R targeting gene KF548063.1 (*SIX*8a) resulted in 400 bp amplicon only with *Foc* R4 VCGs of 0120, 01213, 01216, and 01213/16 (Figure 2B). Of the 31 primer sets tested for *Foc* TR4, *Foc*TR4F/*Foc*TR4R primer targeted to gene KX434998.1 (*SIX*1) generated 250 bp amplicon with *Foc* TR4 VCGs of 01216, 01213, and 01213/16 (Figure 2C). Out of a four primer set designed for STR4, *Foc*STR4F/*Foc*STR4R targeting KX434398.1 (*SIX*7) gene resulted in the generation of 250 bp amplicon with *Foc* STR4 VCG 0120 (Figure 2D).

### 3.2. Sensitivity of the Markers

The gel showing PCR amplification products demonstrated that the lowest successful amplification was observed at 1.0 pg µL^−1^ and 0.1 pg µL^−1^ of genomic DNA on *Foc* R1 and R4, respectively (Figure 3A,B), and the optimal annealing temperature of the race-specific primer ranged between 58 to 66 °C depending on the race (Table 2).

### 3.3. In Planta Detection

The inoculation of *Foc* isolates of R1, STR4, and TR4 on the cv. Grande Naine resulted in the expression of typical external symptoms of leaf yellowing after 18–20 days of inoculation (Figure 4). All the *Foc* races inoculated tissue-cultured banana plants exhibited severe wilting and internal vascular discoloration in both the corm and pseudostem at 25–30 days after inoculation (Figure 4A,C). The study observed no differences in symptoms between the *Foc* races inoculated plants in terms of the number of days required for symptom expression or severity (Figure 4B,D). The tested *Foc* races were successfully recovered from rhizomes using Komada’s medium (Figure 4E,F) and were further confirmed by the spore morphology (Figure 4G) and molecular method using race-specific primers. Genomic DNA (50 ng) isolated from corm samples from infected plants was used for PCR amplification and successfully generated the respective amplicon using the respective primer set designed for the *Foc* races (*Foc* R1 and *Foc* R4, TR4 and STR4) diagnostics.

## 4. Discussion

The purpose of this study was to develop PCR-based molecular markers that can be used to accurately identify Indian *Foc* R1 and R4 which differ in genome size and SIX organization [42] with other reference *Foc* of the same races. Molecular methods based on Foc race identification have been extensively investigated by several researchers such as Lin et al. [21], Fourie et al. [50], Dita et al. [19], Carvalhais et al. [40], and Magdama [51]. However, the present study used molecular markers that were already available as a positive check [19,40,44], which failed to amplify the Indian *Foc* isolates due to genome variations [42]. There are no race-specific markers specifically available for detecting any of the Indian *Foc* races. Therefore, the study aimed to develop molecular markers that target the effector gene for the accurate detection of Indian *Foc* pathogens infecting banana crops [52,53]. This study presumes that PCR-based molecular markers are relatively affordable, robust, repeatable, trustworthy, and effective for the early detection of *Foc* in bananas.

A range of molecular diagnostic methods has previously been proposed to distinguish races of *Foc* [40]. For instance, Lin et al. [21], Fourie et al. [50], Dita et al. [19], and Magdama [51] demonstrated the use of molecular markers. However, the findings were found to have a major drawback in validating the *Foc* races which have been described elsewhere by Carvalhais et al. [40]. To address these issues, this study utilized effector genes as targets for a molecular diagnostic tool for *Foc* races, as each race of Indian VCGs has unique effector genes [42].

Hypothetical protein XM_018394505.1 of *Fol*4287 is present in Chr-1, 3, 6 and 15 of *Foc* R1 VCG0124 has 3036 bp length which ranged from 1525,130 to 1528165 nt in the genome, where the race-specific primer was targeted in Chr-15 between 875,832 to 875,813 and 875,510 to 875,529 of ~320 bp, respectively, for forward and reverse primers (Appendix A). A comparison of different VCGs revealed that VCG0124 exhibits polymorphism at 320 nt, as shown by multiple sequence analysis (Appendix A). The unique SNP allows race-specific identification of Indian *Foc* R1 which includes VCG 0124, 0215, and 01220. Carvalhais et al. [40] previously reported *SIX*6 to diagnose *Foc* R1 VCGs; however, the primer developed did not show the expected amplification product for five *Foc* R1 isolates, including VCGs 0124 and 0123. Therefore, this study believes that the newly developed primer has paramount importance in diagnosing the Indian *Foc* R1 as it is widely distributed in all the sampled states, especially in Cavendish bananas (Figure 1).

For the detection of *Foc* R4, Lin et al. [21] proposed an RAPD primer to diagnose *Foc* R4 VCGs. However, Dita et al. [19] and Magdama, [51] reported that the primers are cross-reactive with *Foc* R1 and nonpathogenic *Foc*. Therefore, in this study, a set of primers *Foc*R4F/*Foc*R4R that has an amplicon size of ~400 bp was designed to target both TR4 and STR4 isolates based on *SIX*8a which is present in Chr-14. Commonly, *SIX*8a is present in the *Foc* STR4 and TR4 which is similar to KF548063.1 of *Fol*4287 and has 713 bp in length from where the primer was designed in 483–502 nt and 661–680 nt, respectively for forward and reverse primer (Appendix A). From Figure 2B, the primer was found to be specific to Indian *Foc* R4 isolates that do not have any cross-reactions. This result was further supported by the results of Carvalhais et al. [40], Fraser-Smith et al. [54], and Czislowski et al. [34], who reported that *SIX*8a is present in both TR4 and STR4, and *SIX*8b is specific to STR4 and can be utilized for molecular diagnosis of *Foc* R4 isolates of divergent origin. In contrast to the above, our previous whole-genome study revealed that *SIX*8b is present in both the Indian races of *Foc* R4 [42]. Therefore, the newly developed *Foc*R4F/*Foc*R4R primer containing both conserved and differential regions in 483–502 nt and 661–680 nt, respectively (Appendix A) were utilized for effective diagnosis of *Foc* R4 isolates of the study area.

For the specific identification of *Foc* TR4, *SIX*1a_266 primer has been effectively utilized by Carvalhais et al. [40], but with a restriction digestion method. In this method, primers flanked a recognition site of the enzyme HpyAV, which is only present in the *SIX*1 gene homolog “a”, which is unique to TR4 of VCG 01213/16 was used for diagnosis [34]. To simplify the method, *Foc*TR4F/*Foc*TR4R primer was designed to identify the Indian *Foc* TR4 which has 17% polymorphism with gaps (Appendix A). The *SIX*1a gene was encoded in the CM028841.1 of Chr-14 which ranged between 1,474,171 and 1,475,025 with a total of 851 bp nt. However, the actual sequence length of *SIX*1a used by Carvalhais et al. [40] was 815 bp nt which is 83% similar to Indian *Foc* TR4 (Appendix A). Moreover, the *SIX*1a primer of Carvalhais et al. [40] included the homologous of ‘a’ to ‘g’, yet the *Foc* TR4 of Indian VCG01213/16 has a single homologous copy. From Figure 2C, it was observed that the primer was proved to be repeatable, robust, and specific to the amplicon size ~250 bp, where the forward primer binding was at 162–181 nt and reverse primer binding at 366–383 nt (Appendix A).

Based on the report of Czislowski et al. [40], the present study makes use of *SIX*7a for the identification of STR4 as it is found to be specific to this race with a wide range of VCGs such as 0120, 0120/15, 0126, 0129, 01211, and 01215. We previously reported that the genome of *Foc* STR4 VCG0120 has a total of 14 homologous copies of *SIX*8 across the different chromosomes. Therefore, using *SIX*8 based primer for the specific detection of Indian *Foc* STR4 remains a matter of risk as it has a polymorphic site in the primers based on homologous variants as reported by Carvalhais et al. [40]. Gene KM503196.1 encoding *SIX*7 in *Foc* STR4 having 663 bp nt was selected as a template to design the primer, where the forward primer binding at 310–329 nt and reverse primer binding at 535–554 nt with the amplicon size of ~248 bp (Appendix A). The gene was present in the Chr-14 (CM028826.1) of Indian *Foc* STR4 between 1,288,728 and 1,289,395 nt (Appendix A). From Figure 2D, the study found that the newly developed primer *Foc*STR4F/*Foc*STR4R was robust and specific.

Based on the aforementioned findings, the study corroborates the results of Fraser-Smith et al. [54], Rocha et al. [53], Ayukawa et al. [55], who developed molecular markers for the effective detection of *Foc* races. As the developed molecular markers allowed the identification of a strain of *Foc* R1 and R4, these race-specific markers can identify *Foc* races through conventional PCR within a few hours whereas pathogenicity testing using differential cultivars requires several months for race determination [20]. Thus, time-bound remedies to standing banana can be implemented immediately rather than taking several months for the identification to manage this lethal disease. Maymon et al. [56] recently used primers developed by O’Neill et al. [57] for recognizing *Foc* races in Israel, which were originally developed for *Foc* strains of northern Queensland. Similarly, the newly developed markers of this study can be used to identify these emerging *Foc* races with larger genome sizes and diverse SIX gene profiles, besides sensitizing the Indian *Foc* isolates in other parts of the nation or world.

After race typing, isolates were rearranged and assessed based on their geographic state of origin. For this, the races of respective isolates were identified and all the states were combined. Out of 61 samples subjected, a maximum of 68.9% are R1 (44) followed by 31.1% are R4 (19), in which 6.6% are STR4 (4) and 24.6% are TR4 (15). Of the eight states surveyed, *Foc* R1 has been the dominant race present in all the states of India (Figure 1). However, the *Foc* TR4 and STR4 are identified in two states where *Foc* TR4 was distributed in Uttar Pradesh and Bihar while *Foc* STR4 was identified from Gujarat and Madhya Pradesh. The locations and samples were randomly selected during our survey. *Foc* isolates differed in percentages from previously reported studies in which race 1 had been the most common race, instead of other races used in this study [33]. The markers used in this present study should enhance the speed and accuracy of the existing diagnostic ability for *Foc* and provide a jumping-off point to investigate similar regions involved in pathogenicity and race development for various researchers. This study exploited the variation within the whole-genome sequences of *Foc* races for developing *Foc* races (R1, R4, TR4, and ST4) specific markers.

## 5. Conclusions

To conclude, the present study has resulted in the development of *Foc* races’ specific molecular markers targeting the effector gene for the accurate diagnosis and detection of Indian *Foc* R1, R4, TR4, and STR4 strains affecting a wide range of banana cultivars including the Cavendish group of bananas which forms the major exporting variety in the world including India. These markers were extensively tested with a large number of isolates of each *Foc* race collected from different geographical locations of different states of India to prove their specificity and robustness in diagnosing all the *Foc* races infecting bananas. Moreover, these markers were also proved to be useful in in planta detection and also with up to 0.025 pg µL^−1^ levels of *Foc* DNA present in the sample. Therefore, the markers developed would be useful not only for quarantine purposes worldwide but also in designing and developing effective and sustainable management practices for this destructive disease.

## Figures and Tables

**Figure 1 jof-08-00053-f001:**
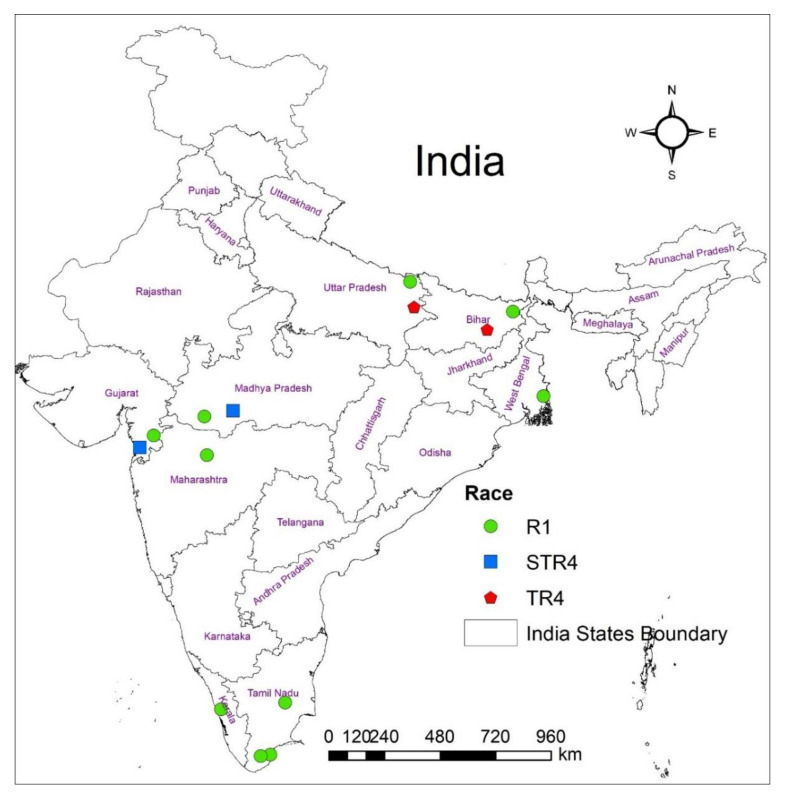
Distribution of *Foc* races across the different states of India. Color and shape denote *Foc* races infecting Cavendish banana.

**Figure 2 jof-08-00053-f002:**
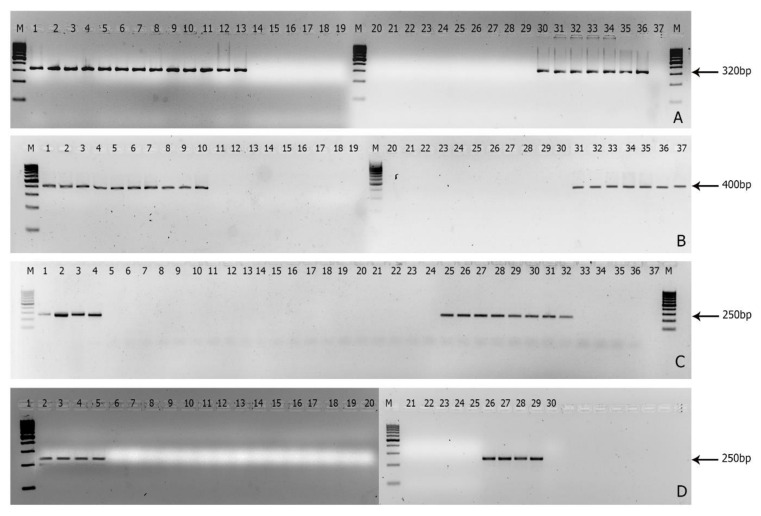
Development of molecular markers for the specific detection of *Foc* races; (**A**): amplicons of *Foc*R1F/*Foc*R1R primer set for *Foc* R1; (**B**): amplicons of *Foc*R4F/*Foc*R4R primer set for *Foc* R4; (**C**): amplicons of *Foc*TR4F/*Foc*TR4R primer set for *Foc* TR4; and (**D**): amplicons of *Foc*STR4F/*Foc*STR4R primer set for *Foc* STR4. For A: M: Marker; Lane 1–13: *Foc* R1 VCGs; Lane 14–20: *Foc* VCGs of TR4 and STR4 (VCG 01213, 01216, 01213/16, 0120, 0121, 0129, and 01211); Lane 21: *Pseudocercospora eumusae*; Lane 22: *Colletotrichum musae*; Lane 23: *Klebsiella variicola*; Lane 24–28: Nonpathogenic *Fusarium oxysporum* (np*Fo*) isolates; Lane 29: *Fusarium solani*; and Lane 30–36; *Foc* VCGs of Race 1 (VCGs 0124, 01220, 0125, 0124/5, 0123, 0126 and 01210; and Lane 37; no template control. For B: M: Marker; Lane 1–10: *Foc* race 4 isolates (both TR4 and STR4); Lane 11–17: *Foc* VCGs of R1 (VCG0124, 01220, 0125, 0124/5, 0123, 0126 and 01210); Lane 18–22: np*Fo* isolates; Lane 23: *Pseudocercospora eumusae*; Lane 24: *Colletotrichum musae*; Lane 25: *Klebsiella varicola*; Lane 26: *Fusarium solani*; Lane 27–30: no template control; Lane 31–37: *Foc* R4 VCGs (01213, 01216, 01213/16, 0120, 0121, 0129 and 01211); For C: M: Marker; Lane 1–4: *Foc* VCG 01213, 01216, 01213 and 01213/16; Lane 5–12: *Foc* R1 and STR4 VCGs (VCG 0124, 01220, 0125, 0124/5, 0123, 0126 and 01210 and 0120); Lane 13–17: np*Fo* isolates; Lane 18: *F. solani*; Lane 19: *Pseudocercospora eumusae*; Lane 20: *Colletotrichum musae*; Lane 21: *Klebsiella variicola*; Lane 22 and 23 np*Fo* isolates Lane 24–32; *Foc* TR4 isolates; Lane 33–36: *Foc* isolates of R1 and STR4; and Lane 37: no template control. For D: Lane 1: Marker; Lane 2–5: *Foc* STR4 isolates; Lane 6–16: *Foc* VCGs of TR4 and R1 (VCG01213, 01216, 01213/16, 0124, 01220, 0125, 0124/5, 0123, 0126, 01210 and 0128); Lane 17–20; np*Fo* isolates; Lane 21: *Fusarium solani*; Lane 22: *Pseudocercospora eumusae*; Lane 23: *Colletotrichum musae*; Lane 24: *Klebsiella varicola*; Lane 25–28: *Foc* VCGs of STR4 (0120, 0121, 0129 and 01211); and Lane 29–30: no template control.

**Figure 3 jof-08-00053-f003:**
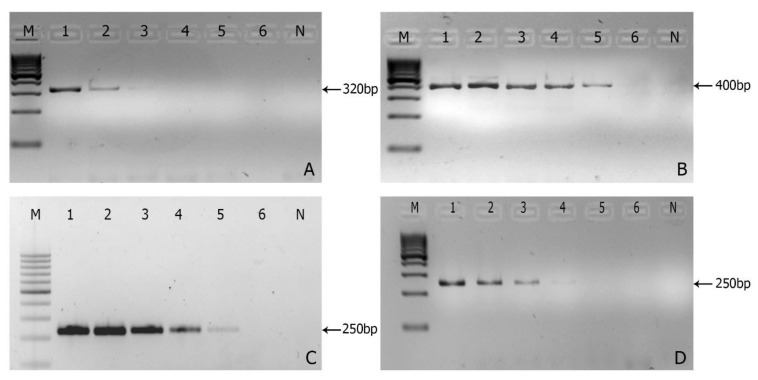
Sensitivity of the primer sets *FocR1F/FocR1R* (**A**), *FocR4F/FocR4R* (**B**), *Foc*TR4F*/Foc*TR4R (**C**), and *Foc*STR4F*/Foc*STR4R (**D**). A serial dilution of genomic DNA ranging from 200 ng to 0.02 pg µL^−1^ was used; where 1: 200 ng; 2: 100 ng; 3: 10 ng; 4: 1 ng; 5: 10 pg; and 6: 1 pg. Lanes M—100 bp Marker; N—Negative control.

**Figure 4 jof-08-00053-f004:**
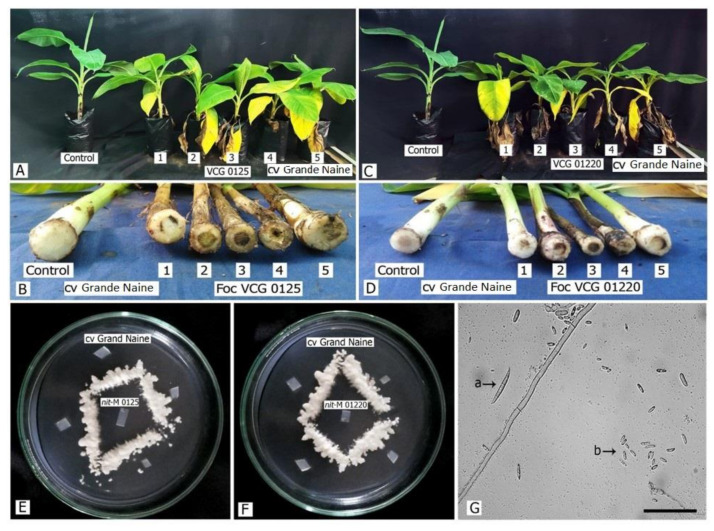
Banana cv. Grande Naine 25–30 days after inoculation with different races of *Foc* isolates. (**A**,**C**): Pathogenicity test of VCG0125 and VCG01220; (**B**,**D**): Cross-section of rhizome for vascular discoloration; (**E**,**F**): isolation and confirmation of VCGs by nit-M testers; and (**G**): Spore morphology of 7 days old culture of *Foc* VCG 0125 where a: macroconidia and b: microconidia, the scale bar is 100 µm and the image taken under 40× magnification on the Nikon^®^ light microscope Eclipse TE2000.

**Table 1 jof-08-00053-t001:** Details of *Foc* isolates used in the study, code, host genome, geographical origin, and their race/vegetative compatibility group (VCG) characterization and pathogenicity on the cv. Grande Naine.

S. No.	Isolate	Cultivar (Genome Status)	Geographical Locations	Race	VCG ^†^	Pathogenicity on cv. Grande Naine (AAA)	PCR Diagnosis for			
							TR4	STR4	R4	R1
1	SU1	Grande Naine (AAA)	Karjan, Gujarat	1	01220	+	-	-	-	+
2	SU2	Grande Naine (AAA)	Karjan, Gujarat	1	01220	+	-	-	-	+
3	SU3	Grande Naine (AAA)	Surat, Gujarat	4	0120	+	-	+	+	-
4	SU4	Grande Naine (AAA)	Karjan, Gujarat	1	01220	+	-	-	-	+
5	SU5	Grande Naine (AAA)	Karjan, Gujarat	1	01220	+	-	-	-	+
6	UP1	Grande Naine (AAA)	Gurlirumkadava, UP	1	0125	+	-	-	-	+
7	UP2	Grande Naine (AAA)	Pilavagam, UP	1	01220	+	-	-	-	+
8	UP3	Grande Naine (AAA)	Kaleelnagar, UP	1	0125	+	-	-	-	+
9	UP4	Grande Naine (AAA)	Puphyhee, UP	4	01216	+	+	-	+	-
10	UP5	Grande Naine (AAA)	Sabeyadala, UP	1	01220	+	-	-	-	+
11	UP6	Grande Naine (AAA)	Buchiyang, UP	1	0125	+	-	-	-	+
12	UP7	Grande Naine (AAA)	Siswabazar,UP	4	01216	+	+	-	+	-
13	UP8	Grande Naine (AAA)	Siswabazar,UP	4	01216	+	+	-	+	-
14	UP9	Grande Naine (AAA)	Siswabazar,UP	4	01216	+	+	-	+	-
15	UP10	Grande Naine (AAA)	Siswabazar,UP	4	01216	+	+	-	+	-
16	UP11	Grande Naine (AAA)	Siswabazar,UP	4	01216	+	+	-	+	-
17	B1	Robusta (AAA)	Kattihar, Bihar	4	01213/16	+	+	-	+	-
18	B2	Robusta (AAA)	Kattihar, Bihar	4	01213/16	+	+	-	+	-
19	B3	Robusta (AAA)	Kattihar, Bihar	4	01213/16	+	+	-	+	-
20	B4	Grande Naine (AAA)	Kattihar, Bihar	4	01213/16	+	+	-	+	-
21	B5	Grande Naine (AAA)	Katihar, Bihar	4	01213/16	+	+	-	+	-
22	B6	Monthan (AAA)	Katihar, Bihar	1	01220	+	-	-	-	+
23	B7	Monthan (AAA)	Katihar, Bihar	1	01220	+	-	-	-	+
24	B8	Monthan (AAA)	Katihar, Bihar	1	01220	+	-	-	-	+
25	B9	Malbhog (AAA)	Katihar Bihar	1	0125	+	-	-	-	+
26	B10	Grande Naine (AAA)	Katihar Bihar	4	01216	+	+	-	+	-
27	B11	Grande Naine (AAA)	Katihar, Bihar	4	01216	+	+	-	+	-
28	B12	Grande Naine (AAA)	Purnia, Bihar	4	01216	+	+	-	+	-
29	B13	Grande Naine (AAA)	Purnia, Bihar	4	01216	+	+	-	+	-
30	TN1	Grande Naine (AAA)	Theni, TN	1	0124	+	-	-	-	+
31	TN2	Grande Naine (AAA)	Theni, TN	1	0125	+	-	-	-	+
32	TN3	Robuta (AAA)	Theni, TN	1	01220	+	-	-	-	+
33	TN4	Grande Naine (AAA)	Theni, TN	1	0125	+	-	-	-	+
34	TN5	Papoulou (AAB)	Theni, TN	1	01220	+	-	-	-	+
35	TN6	Grande Naine (AAA)	Theni, TN	1	0125	+	-	-	-	+
36	TN7	Grande Naine (AAA)	Theni, TN	1	0125	+	-	-	-	+
37	TN8	Grande Naine (AAA)	Theni, TN	1	0125	+	-	-	-	+
38	TN9	Grande Naine (AAA)	Theni, TN	1	0124	+	-	-	-	+
39	TN10	Grande Naine (AAA)	Theni, TN	1	0125	+	-	-	-	+
40	TN11	Grande Naine (AAA)	Theni, TN	1	0125	+	-	-	-	+
41	TN12	Grande Naine (AAA)	Theni, TN	1	0124	+	-	-	-	+
42	TN13	Grande Naine (AAA)	Theni. TN	1	01220	+	-	-	-	+
43	TN14	Grande Naine (AAA)	Theni, TN	1	0125	+	-	-	-	+
44	TN15	Karpuravalli (ABB)	Kattuputhur, TN	1	0125	+	-	-	-	+
45	TN16	Rasthali (AAB)	Namakkal, TN	1	0125	+	-	-	-	+
46	TN17	Rasthali (AAB)	Tirunelveli, TN	1	0125	+	-	-	-	+
47	TN18	Monthan (ABB)	Tuticorin, TN	1	01220	+	-	-	-	+
48	TN19	Ney Poovan (AB)	Tuticorin, TN	1	01220	+	-	-	-	+
49	TN20	Grande Naine (AAA)	Coimbatore, TN	1	*	+	-	-	-	+
50	TN21	Karpuravalli (ABB)	Coimbatore, TN	1	01220	+	-	-	-	+
51	TN22	Rasthali (AAB)	Coimbatore, TN	1	*	+	-	-	-	+
52	TN23	Ney Poovan (AB)	Coimbatore, TN	1	*	+	-	-	-	+
53	Ke1	Rasthali (AAB)	Thrissur, Kerala	1	*	+	-	-	-	+
54	Ke2	Rasthali (AAB)	Thrissur, Kerala	1	*	+	-	-	-	+
55	Ke3	Rasthali (AAB)	Thrissur, Kerala	1	*	+	-	-	-	+
56	Ke4	Big Ebanga (AAB)	Thrissur, Kerala	1	0125	+	-	-	-	+
57	Ke5	Rasthali (AAB)	Thrissur, Kerala	1	*	+	-	-	-	+
58	MP1	Grande Naine (AAA)	Burhanpur, MP	4	0120	+	-	+	+	-
59	MP2	Grande Naine (AAA)	Burhanpur, MP	4	0120	+	-	+	+	-
60	MP3	Grande Naine (AAA)	Burhanpur, MP	4	0120	+	-	+	+	-
61	MP4	Grande Naine (AAA)	Burhanpur, MP	1	*	+	-	-	-	+

^†^ Race designation according to newly developed molecular markers of this study and VCGs by *nit*-M testers. * Isolates are not complemented with any of the respective races, it is considered as a race of unknown VCG.

**Table 2 jof-08-00053-t002:** List of *Foc* race-specific primers used in this study.

Primer Name	Gene ID	Primer sequence (5′ to 3′)	Length (bp)	Annealing Conditions
*Foc*R1F *Foc*R1R	XM_018394505.1 (Hypothetical protein)	TACCTCCTTGGTCGACAGGT CAGACTTCCAACGTCTCGGT	320 bp	62 °C for 30 s 30 cycles
*Foc*R4F *Foc*R4R	KF548063.1 (*SIX*8a)	CGCACTCTTACGTTGAGGAT TCCACGCAACACTAGCTACT	400 bp	66 °C for 30 s 30 cycles
*Foc*TR4F *Foc*TR4R	KX434998.1 (*SIX*1a)	TGATTTGCCGTGGAATGACA TGGTCTTGACACGACCCA	250 bp	65 °C for 30 s 30 cycles
*Foc*STR4F *Foc*STR4R	KM503196.1 (*SIX*7a)	GCGCAAGTAGTCTTGCTTCC ATTAAGCGGTTGGCGTATTG	250 bp	58 °C for 30 s 30 cycles

## Data Availability

The assembled short-read genome sequences of *Foc* Race1 (VCG0124), *Foc* STR4 (VCG0120), and *Foc* TR4 (VCG 01213/16) infecting Cavendish group banana have been deposited at DDBJ/EMBL/GenBank under accession number GCA_011316005.3, GCA_016802205.1, and GCA_014282265.3, respectively. All the data generated or analyzed during this study are included in this published article.

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
