# Peer review of "Development of PCR-Based Race-Specific Markers for Differentiation of Indian Fusarium oxysporum f. sp. cubense, the Causal Agent of Fusarium Wilt in Banana"

_jof, 2022, doi:10.3390/jof8010053_

Round 1

Reviewer 1 Report

The paper “Development of PCR based race-specific markers for differentiation of Indian Fusarium oxysporum f. sp. cubense, the Causal agent of Fusarium wilt in Banana” by Thangavelu et al. assessed a set of molecular markers for specific detection of Foc race, focusing on isolates collected from India. Considering that investigation of the economically important plant pathogens is of utmost interest to evaluations of plant health, the current study contribution is pertinent and needed, especially in the region.

The research is pertinent, unfortunately, the paper shows several issues that need to be fully addressed before it can be published. I have some comments and suggestions for the manuscript, as indicated below.

Introduction.

L109-111 Please, include a reference

L111-113 The sentence reads strange, and not in accordance with reality, given that there are currently an important number of molecular diagnostic assays (PCR, qPCR, LAMP etc) which are precise and reproducible. Maybe you mean that molecular markers more precise are required? I would try to rephrase it.

L114-117 An important number of molecular markers for Foc TR4 detection is already available, even molecular markers corresponding to avirulence genes, such as SIX genes and W2789 (Carvalhais 2019, Li 2013). So, it is not completely true that… The recently developed molecular diagnostics methods were generally based on core genomic regions…

Additionally, I would explain more about all molecular markers for Foc TR4 detection that have been recently developed.

L118-119 Which other strains, please specify.

L124 Please, be consistent with the name of genes that should be always in italics (SIX genes)

L129-131 Do you have bibliographic a reference for this sentence

Materials and Methods

L151-152 Please, could you provide a supplemental table with the 23 Foc isolates, indicating VCGs  

One of the most interesting aspect of this work is the comparison of whole genome sequences of R1, TR4 and STR4, as indicated in L21-23 (abstract). However, in section 2.3 (Sequence alignment and primer design) there is a poor explanation on how this whole genome comparison was conducted. Currently, there are more than 20 whole genome sequences of Foc TR4 publicly available and for R1…did you use these genomes? If not why?

L192-194 this explanation is not clear…please, specify what targeted gene candidates    

Additionally, the methodology to select specific regions needs a much more thorough explanation.

L209-211 Please, could you provide a figure indicating these variable regions and highly conserved regions?

L219-220 Why did you run just 20 cycles of amplification? What happens if PCR cycles are increased to 30-35?

Results

The results need more detail, for example, the whole genome comparison and the screening for regions specific to each race was not presented. Results section should be written to describe the significance of the results based on published literature, including Thangavelu 2021. J Fungi. 7, 717.

L301-303 What you mean with the sentence separate assays? DNA extracted from different samples?  Different PCRs on the same DNA extractions? Please clarify.

Figure 3. It is not possible identify/differentiate Foc TR4 isloates fo India from those reference Foc isolates. Please, indicate it.

In my opinion, PCRs with molecular markers were performed with a low number of Foc isolates (figure 3). Did you also run the PCRs for all isolates indicated in table 1 and the 23 reference Foc isolates?

Discussion

L344-349 Comparison of SIX genes is discussed but no presented in results section.

With these changes, this research will help to the development of Foc races specific molecular markers for Foc isolates from India.

Author Response

Reply to the reviewers’ comments of jof-1510505

The authors’ answers are in black and reviewer questions and comments are in blue.

Reviewer-1

Introduction

L109-111 Please, include a reference

The reference has been included

L111-113 The sentence reads strange, and not in accordance with reality, given that there are currently an important number of molecular diagnostic assays (PCR, qPCR, LAMP etc) which are precise and reproducible. Maybe you mean that molecular markers more precise are required? I would try to rephrase it.

Corrected as suggested

L114-117 An important number of molecular markers for Foc TR4 detection is already available, even molecular markers corresponding to avirulence genes, such as SIX genes and W2789 (Carvalhais 2019, Li 2013). So, it is not completely true that… The recently developed molecular diagnostics methods were generally based on core genomic regions…

Additionally, I would explain more about all molecular markers for Foc TR4 detection that have been recently developed.

The sentence has been rephrased in L114-116

L118-119 Which other strains, please specify.

Its not strains, actually VCGs related to Foc TR4

L124 Please, be consistent with the name of genes that should be always in italics (SIX genes

Corrected throughout the manuscript

L129-131 Do you have bibliographic a reference for this sentence

This is our personnel observation, the same has been included in the manuscript

Materials and Methods

L151-152 Please, could you provide a supplemental table with the 23 Foc isolates, indicating VCGs.

This has been included as a Supplementary table in the revised version of the manuscript.

One of the most interesting aspects of this work is the comparison of whole genome sequences of R1, TR4 and STR4, as indicated in L21-23 (abstract). However, in section 2.3 (Sequence alignment and primer design) there is a poor explanation on how this whole genome comparison was conducted. Currently, there are more than 20 whole genome sequences of Foc TR4 publicly available and for R1…did you use these genomes? If not why?

The study focused only on the genomes of Indian isolates that have been submitted by us which was furnished in the data availability statement. To elucidate the difference between the Indian isolates with the reference Fol4287 (ASM14995v2; txid426428) and Foc TR4 (ASM799451v1; txid61366) Interactive Genomics Viewer was used. As primers are targeted to effector genes, the method of identification and primer design are comprehended in the materials and method section.

L192-194 this explanation is not clear…please, specify what targeted gene candidates. Additionally, the methodology to select specific regions needs a much more thorough explanation.

Further explanation on screening and targeting the primers was included in the revised manuscript.

L209-211 Please, could you provide a figure indicating these variable regions and highly conserved regions?

The conserved and variable regions where forward and reverse primer binding sites were already depicted in the supplementary figures 1-4.

L219-220 Why did you run just 20 cycles of amplification? What happens if PCR cycles are increased to 30-35?

It is 30 cycles (Typos) and the same is corrected

Results

The results need more detail, for example, the whole genome comparison and the screening for regions specific to each race was not presented. Results section should be written to describe the significance of the results based on published literature, including Thangavelu 2021. J Fungi. 7, 717.

A brief report of the same from Thangavelu et al., 2021 has been furnished in the result section.

L301-303 What you mean with the sentence separate assays? DNA extracted from different samples?  Different PCRs on the same DNA extractions? Please clarify.

The sentence has been rephrased into “Reproducibility of PCR assay was further confirmed by two different operators, on three different occasions, with the same set of samples”. Therefore, separate means to the PCR reaction done by two different professionals of different PCR machines in various departments of the Institute at different times with suitable technical replications.

Figure 3. It is not possible to identify/differentiate Foc TR4 isolates of India from those reference Foc isolates. Please, indicate it.

In my opinion, PCRs with molecular markers were performed with a low number of Foc isolates (figure 3). Did you also run the PCRs for all isolates indicated in table 1 and the 23 reference Foc isolates?

We run the PCR markers for all Indian isolates which were furnished in the table, besides we used 23 reference isolates of different VCGs. However, for the presentation, we included only representative Indian isolates which also includes false-positive results with existing molecular markers.

Discussion

L344-349 Comparison of SIX genes is discussed but not presented in results section.

The presence and distribution of the SIX genes in the Indian Foc isolates have been already extensively described by Thangavelu et al., 2021 J Fungi. 7, 717. However, the purpose of the study is not to describe the SIX genes, therefore the authors are limited to providing the information related to the genes used for the development of molecular markers.

With these changes, this research will help to the development of Foc races specific molecular markers for Foc isolates from India.

The authors extend their gratitude to the reviewer for his constructive comments and suggestions to improve the core concept and understanding.

Reviewer 2 Report

This manuscript communicates the development of Foc race specific primers that will help to cut down the time in diagnosis. Current methods require multiple time consuming and often costly steps of diagnosis, leave alone the challenge of specificity. The results of the study are thus very relevant for the management of the disease. More still, I see the approaches used in the study to be relevant for addressing similar constraints with the disease elsewhere or other diseases.

However, minor edits are proposed for improvement of the manuscript as described below.

  • Some content relevant to the materials and methods section added in the results section e.g. in L 256-259, and L294-295. Please integrate these in the MM section - seed attached pdf for detailed comments.
  • Some results presented in the discussion section- see the attached pdf for the detailed comments
  • Objective 1 of the study not explicitly answered in the results. Organizing of the MM, results and discussions along the stipulated objectives will help improve the flow of this good work.
  • Additional edits and comments are provided within the manuscript -see the attached pdf 

Author Response

Reply to the reviewers’ comments of jof-1510505

The authors’ answers are in black and reviewer questions and comments are in blue.

Reviewer-2

Some content relevant to the materials and methods section added in the results section e.g. in L 256-259, and L294-295. Please integrate these in the MM section - see attached pdf for detailed comments. Some results presented in the discussion section- see the attached pdf for the detailed comments.

The manuscript has been revised in accordance with the reviewer suggestion

Objective 1 of the study not explicitly answered in the results.

The same concern has also been raised by one of the reviewers, and the results of objective 1 have been included in the result section in brief.

Organizing of the MM, results and discussions along the stipulated objectives will help improve the flow of this good work.

MM has been rearranged according to the reviewer suggestion (and authors are grateful for this good comment of the reviewer).

Additional edits and comments are provided within the manuscript -see the attached pdf

All the comments and suggestions recommended by the reviewer in the PDF have been included in the revised manuscript and we are thankful for the extensive review done which has improved the readability and understanding of the MS.

Reviewer 3 Report

The manuscript entitled by Development of PCR based race-specific markers for differentiation of Indian Fusarium oxysporum f. sp. cubense, the Causal agent of Fusarium wilt in Banana provided a novel molecular marker to distinguish Foc TR1 and Foc TR4. The study is really interesting, and writted well. However, many information have been reported on the Foc diagnosis, such as  ‘Lin et al, Eur. J. Plant Pathol, 2009, Wang et al, Molecular Biology Reports,2012, Yang et al,Crop Protection,2015’. I think no highlights or innovation is found in the present study. In addition, the below should be checked:

  1. 1. Line 13: "Foc" should not be italicized. Cheek it throughout the manuscript.
  2. Along the text: change "PCR based" to "PCR-based".
  3. Line 39: "148 million tons are produced annually from 135 countries", I have checked the literature "[1]" which you cited, but did not find such expression.
  4.  Line 67-68: please providdata sources.
  5. Line 77: change " of banana field" to " of the banana field".

6.Line 124: "SIX" should be italic, check italic throughout the manuscript.

  1. Line 125-126: "Fo" should not be italicized.
  2. Line 151: the "DNA" is unclear. Does DNA mean genomic DNA?
  3. Line 204: 20℃ or -20℃ ? Please, check.
  4. Line 244: "106conidia g−1",  could you please explain how do you obtain conidia from "sand: maize (19:1) medium"?
  5. The experimental methods of"In planta detection of Foc races" are unclear and lacking sufficient detail to enable a clear understanding of what was done. For instance, you said "planted with 3-month-old tissue-cultured plants", but did not tell how to set up treatments? and how many plants for each treatment? "After 25 days of inoculation", how did you treated the treatments? And how many times? In addition, In the figure 2 A and C, I noticed that the banana seedlings were in tissue culture bags, is it the original soil matrix? "Total genomic DNA from plant tissues", which part of the plant tissue?
  6. The method is inconsistent with the result. For example, for the experiment "In planta detection", you said inoculated 25 days in line 246. But in your results in line 314, you said the results are from 25 to 30 days.
  7. Line 313: it is wrong for "Figure 4".
  8. Line 326: Required more detail of figure 2G. How long have the spores been cultured? What is the value of bar?
  9. The references are full of irregularities and errors. These needs to be extensively and carefully edited.

Author Response

Reply to the reviewers’ comments of jof-1510505

The authors’ answers are in black and reviewer questions and comments are in blue.

Reviwer-3

Line 13: "Foc" should not be italicized. Cheek it throughout the manuscript.

We generally follow italicizing of Foc as per the binomial norms. If the reviewer is particular about not italicizing the Foc we will change it later. 

Along the text: change "PCR based" to "PCR-based".

Corrected as suggested.

Line 39: "148 million tons are produced annually from 135 countries", I have checked the literature "[1]" which you cited, but did not find such expression.

The correct reference is included

Line 67-68: please provide data sources.

Included the reference 

Line 77: change " of banana field" to " of the banana field".

Corrected as suggested.

Line 124: "SIX" should be italic, check italic throughout the manuscript.

Corrected throughout the manuscript as suggested.

Line 125-126: "Fo" should not be italicized.

We generally follow italicizing of Fo as per the binomial norms. If the reviewer is particular about not italicizing the Fo we will change it later. 

Line 151: the "DNA" is unclear. Does DNA mean genomic DNA?

The sentence is rephrased into “Further, the genomic DNA of 23 reference Foc isolates (Supplementary Table 1) of different VCGs (obtained from QDPI & F, Brisbane, Australia)”.

Line 204: 20℃ or -20℃? Please, check.

It is -20℃. The typo was corrected

Line 244: "106 conidia g−1", could you please explain how do you obtain conidia from "sand: maize (19:1) medium"?

A loop full of actively growing PDA Foc cultures of different races were inoculated with sand: maize medium (19:1) and waited until to reach culture to cover the entire area of the medium. The spore count was done fifteen days after inoculation by serial dilution which has been used for treating banana plantlets.

The experimental methods of "In planta detection of Foc races" are unclear and lacking sufficient detail to enable a clear understanding of what was done. For instance, you said "planted with 3-month-old tissue-cultured plants", but did not tell how to set up treatments? and how many plants for each treatment? "After 25 days of inoculation", how did you treated the treatments? And how many times? In addition, In the figure 2 A and C, I noticed that the banana seedlings were in tissue culture bags, is it the original soil matrix? "Total genomic DNA from plant tissues", which part of the plant tissue?

Section 2.6 has been completely rewritten with more information which includes configuration of poly grow bag and filling mixture, experimental setup and conditions besides methods of treatment and sampling procedure. We used discoloured vascular strands of the corm for establishing the culture and DNA isolation.

The method is inconsistent with the result. For example, for the experiment "In planta detection", you said inoculated 25 days in line 246. But in your results in line 314, you said the results are from 25 to 30 days.

The initial observation was done after 25 days of inoculation which was mentioned in MM, however, the wilting was observed between 25-30 days after inoculation in different replications.

Line 313: it is wrong for "Figure 4".

Typo corrected into Figure 2

Line 326: Required more detail of figure 2G. How long have the spores been cultured? What is the value of bar?

Detailed description is given in the footnoted of the figure as suggested.

The references are full of irregularities and errors. These needs to be extensively and carefully edited.

We are using Mendeley software for reference and citation, however, as suggested by the reviewer we carefully revised according to the JoF format.

Reviewer 4 Report

I think the article is interesting and that the molecular tools to diagnose the Foc strains available up to now were insufficient. This work provides an effective new tool but it is not clear if it is useful only for the geographical region of India, or if it would be valid for all the areas in which Foc is present. In general, a more in-depth review of molecular tools for Foc should be made in the introduction and their ineffectiveness should be discussed further in the discussion section.

In addition, I have some questions and suggestions that could improve the manuscript.

Abstract

Line 17: replace Grand Naine by Grand Nain or Grande Naine

Line 21: add representative strains of…

Line 22: Why f. sp. lycopersici was included in this study?

It is not clear how many isolates are compared and if they are from India?

Lines 32-34: Is this statement valid for Foc races worldwide distributed or only for Indian races? Please, clarify this aspect.

Line 42: replace Cavendish group by Cavendish subgroup. Correct this mistake along the manuscript.

Introduction

Line 51-66: please add some references about Foc races. Author could add information about the Race 4 that is not TR4 or STR4.

Line 55: replace Cavendish group by Cavendish subgroup.

Line 76: please, be consistent with capitalization when it comes to the word Cavendish.

Line 81, 83, 109: I think the comma in the number 1220 should be removed

Line 103: It is not the first time VCG is mentioned in the text. Define the first time in Line 57.

Line 106: try to find a synonym o replace differentiate as it is repetitive with differential.

Line 109: What does 'and vice versa' mean here?

Line 118-119: please clarify ‘other strains which are closely related to TR4’.

Lines 119-122: add references for this sentence or remove it. This could be discussing in discussion section.

Line 126: incorrect expression ‘races of Fo’. The f. sp. has races, not species. Please correct this sentence.

Line 129-132: references should be added.

Line 134: change race4 by R4.

Lines 132-135: these lines are a mix between conclusions and objective, please move it to the right place.

Lines 114-138: in the introduction there should be a more in-depth review of the molecular tools used for the diagnosis of Foc races.

Lines 140-142: is goal 2 valid for Foc races of all geographic areas or just for India?

Material and Methods

Line 148: Please, be consistent in your use of numbers. Change five by 5.

Line 152: Please, be consistent in your use of numbers. Change five by 5 and one by 1.

Line 157: Change Grand Naine by Grand Nain or Grande Naine. Add sub to ‘Cavendish AAA group’ (subgroup).

Lines 157-162: I think this information should be in other epigraph.

Line 164: Change Grand Naine by Grand Nain or Grande Naine.

Table 1: Change Grand Naine by Grand Nain or Grande Naine. Change race4 and race1 by R4 and R1. Please specify which molecular markers were used (references), those existing prior to this work or those developed in this article.

Line 167: It is not necessary to describe the complete protocol, only the modifications.

Line 212: Please, be consistent in your use of numbers.

Line 217-217: Please add primers and DNA concentrations.

Line 221: T4 is missing an R.

Table 2: Change FocT4 by FocTR4 and FocS4 by FocSTR4

Line 229: remove 0 in 68.0.

Line 232: add ‘section’ to ‘mentioned in 2.1’.

Line 245: Change Grand Naine by Grand Nain or Grande Naine.

Line 251: How much DNA is used for pcr?

Line 256: ‘T’ must be added to SR4.

Line 258: Authors could add other f.sp. of Fo and other fusarium species in order to test the primers.

Line 263: 01,220. It is an error? I think it should be 01220.

Line 265: 01,216. It is an error? I think it should be 01216.

Figure 3: I think this figure should be divided in four different figures. In the legend, add ‘s’ to marker word. Add a space between 0121,0129 (0121, 0129).

Lines 294-297: Correspond to materials and methods, please remove them.

Line 308: DNA ranging does not correspond to that mentioned above in the text (25 ng μL−1 to 0.025 pg μL−1  line 295).

Line 311: Change Grand Naine by Grand Nain or Grande Naine.

Line 313: Figure 4 appears in the text before Figure 2. Please, check the order of all figures included in the manuscript.

Line 315: be consistent with capital letters Figure 4a&b (line 300) Figure 2A&C

Lines 320-322: these sentences must be in materials and methods section.

Figure 2. Change Grand Naine by Grand Nain or Grande Naine. Also in figure legend.

Lines 332-333: Remove authors names and et al. ‘Lin et al. [21], Fourie et al., [34], Dita et al., 332 [35], and Magdama, [36]’. I think Carvalhais et al. 2019 reference should be added.

Please check all et al: Should it be italicized or not? after et al there must be a dot or dot and comma (et al.,).

Lines 341-343: these sentences correspond to materials and methods section.

Lines 357-361: these sentences must be in results section not in discussion.

Line 367: these data should be shown.

Lines 388: Does it refer to the R4 that are neither TR4 nor STR4? Please show the results.

Line 390: It is effective against R4 in other geographic areas?

Line 399: Change 2018 by 2019 in Carvalhais et al. reference.

Line 415: ‘1,288,728 and 1289395nt’. Please, be consistent writing numbers.

Line 421: Write R1 and R4 instead of race1 and race 4.

References

Check the use of italics for Latin names.

Line 489: remove semicolon and complete the information.

Line 492: Please, add doi (doi: 10.1079/PAVSNNR202015004).

Line 494: remove semicolon and complete the information (FAO food Outlook).

Line 496: Please, add doi (DOI: 10.17660/ActaHortic.2011.897.44).

Line 566: remove capital letter for f. Sp and Cubense (f. sp. cubense is the correct form).

Line 575: remove capital letter for f. Sp and Cubense.

Line 575: remove capital letter for F. Sp and Cubense.

Author Response

Reply to the reviewers’ comments of jof-1510505

The authors’ answers are in black and reviewer questions and comments are in blue.

Reviewer-4

Abstract

Line 17: replace Grand Naine by Grand Nain or Grande Naine

Grand Nain is replaced by Grande Naine throughout the manuscript

Line 21: add representative strains of…

Instead, we have mentioned the exact genome ID of the isolates

Line 22: Why f. sp. lycopersici was included in this study?

It is not clear how many isolates are compared and if they are from India?

As Fol4287 is a reference genome that has been used for making assembly of Foc, Fol was used. Each genome (one) has been currently available in the public database (NCBI) and the same was compared in this study.

Lines 32-34: Is this statement valid for Foc races worldwide distributed or only for Indian races? Please, clarify this aspect.

It is for the Indian Foc races

Line 42: replace Cavendish group by Cavendish subgroup. Correct this mistake along the manuscript.

As it means different varieties belong to different subgroups, we mention here as Cavendish group which would be more appropriate and hence it is retained as such

Introduction

Line 51-66: please add some references about Foc races. Author could add information about the Race 4 that is not TR4 or STR4.

The authors believe that the information provided is adequate.

Line 76: please, be consistent with capitalization when it comes to the word Cavendish.

Corrected

Line 81, 83, 109: I think the comma in the number 1220 should be removed.

Typos corrected

Line 103: It is not the first time VCG is mentioned in the text. Define the first time in Line 57.

Corrected as suggested

Line 106: try to find a synonym to replace differentiate as it is repetitive with differential.

We used the word discriminate

Line 109: What does 'and vice versa' mean here?

The sentence is corrected                                                     

Line 118-119: please clarify ‘other strains which are closely related to TR4’.

According to Ordonez et al., 2015; Czislowski et al., 2017; Mostert et al., 2017, the VCGs such as those affiliated to VCG 0121 and VCG 0122 are closely related to TR4 are also able to infect Cavendish banana. This was included in the manuscript    

Lines 119-122: add references for this sentence or remove it. This could be discussing in discussion section.

Author believes that the sentence is important to justify the concept, therefore, decided to keep it.

Line 126: incorrect expression ‘races of Fo’. The f. sp. has races, not species. Please correct this sentence.

Thanks for the valuable correction, the sentence has been changed into “the VCGs of Foc vary in SIX protein profile”.

Line 129-132: references should be added.

This is the personnel observation from this study used to emphasise the research gap related to Indian isolates. Thus, mentioned as “personnel observation”.

Line 134: change race4 by R4.

Corrected as suggested.

Lines 132-135: these lines are a mix between conclusions and objective, please move it to the right place.

The sentence has been rephrased

Lines 114-138: in the introduction there should be a more in-depth review of the molecular tools used for the diagnosis of Foc races.

Some points have been included in the introduction, however, as we felt that the inclusion of in-depth review in the discussion section is very important, we have not included the same in the introduction section.

Lines 140-142: is goal 2 valid for Foc races of all geographic areas or just for India?

As aforesaid, the markers can be used to Foc VCGs having higher genome size with a different configuration of SIX genes or only Foc of Indian races. The same is mentioned in the MS.

Material and Methods

Line 148: Please, be consistent in your use of numbers. Change five by 5.

Corrected

Line 152: Please, be consistent in your use of numbers. Change five by 5 and one by 1.

Corrected

Line 157: Change Grand Naine by Grand Nain or Grande Naine. Add sub to ‘Cavendish AAA group’ (subgroup).

The authors used Grande Naine and the remaining changes were done according to the suggestion.

Lines 157-162: I think this information should be in other epigraph.

Modified

Line 164: Change Grand Naine by Grand Nain or Grande Naine.

Authors following Grande Naine in the entire manuscript

Table 1: Change Grand Naine by Grand Nain or Grande Naine. Change race4 and race1 by R4 and R1. Please specify which molecular markers were used (references), those existing prior to this work or those developed in this article.

The authors used our own newly developed molecular markers for recognising Foc races, therefore, the reference is not included in the table footnote.

Line 167: It is not necessary to describe the complete protocol, only the modifications.

As authors using the modified protocol of Dellaporta et al. we described modification in the para wherever needed.

Line 212: Please, be consistent in your use of numbers.

Commonly numbers less than 10 used to write so we followed, now we have changed numbers as a numeral. 

Line 217-217: Please add primers and DNA concentrations.

We used 0.5 µM µL-1 of primers and 50 ng µL-1 of DNA in the PCR reaction.

Line 221: T4 is missing an R.

Typo corrected

Table 2: Change FocT4 by FocTR4 and FocS4 by FocSTR4

Changed as suggested

Line 229: remove 0 in 68.0.

Typo corrected

Line 232: add ‘section’ to ‘mentioned in 2.1’.

Modified accordingly

Line 245: Change Grand Naine by Grand Nain or Grande Naine.

Authors following Grande Naine in the entire manuscript

Line 251: How much DNA is used for PCR?

We used 50 ng µL-1 of DNA

Line 256: ‘T’ must be added to SR4.

Typo corrected

Line 258: Authors could add other f.sp. of Fo and other fusarium species in order to test the primers.

The microorganisms of both fungi and bacteria related to Banana are included in the study besides 23 reference strains.

Line 263: 01,220. It is an error? I think it should be 01220.

Typos corrected

Line 265: 01,216. It is an error? I think it should be 01216.

Typos corrected

Figure 3: I think this figure should be divided in to four different figures. In the legend, add ‘s’ to marker word. Add a space between 0121,0129 (0121, 0129).

We want to retain A, B, C and D for the comparison. The other suggestions made are corrected

Lines 294-297: Correspond to materials and methods, please remove them.

As suggested moved to the materials and method section.

Line 308: DNA ranging does not correspond to that mentioned above in the text (25 ng μL−1 to 0.025 pg μL−1 line 295).

Typos corrected

Line 311: Change Grand Naine by Grand Nain or Grande Naine.

Authors following Grande Naine in the entire manuscript

Line 313: Figure 4 appears in the text before Figure 2. Please, check the order of all figures included in the manuscript.

Typos corrected

Line 315: be consistent with capital letters Figure 4a&b (line 300) Figure 2A&C.

Typos corrected

Lines 320-322: these sentences must be in materials and methods section.

The authors believe that the sentence in this place is appropriate to conclude the experiment.

Figure 2. Change Grand Naine by Grand Nain or Grande Naine. Also in figure legend.

Authors following Grande Naine in the entire manuscript

Lines 332-333: Remove authors names and et al. ‘Lin et al. [21], Fourie et al., [34], Dita et al., 332 [35], and Magdama, [36]’. I think Carvalhais et al. 2019 reference should be added.

Please check all et al: Should it be italicized or not? after et al there must be a dot or dot and comma (et al.,).

According to JoF, authors names can be used for citation without italicizing et al. moreover, Carvalhais et al. 2019 citation was added in the sentence.

Lines 341-343: these sentences correspond to materials and methods section.

Moved accordingly

Lines 357-361: these sentences must be in results section not in discussion.

Moved accordingly

Line 367: these data should be shown.

We have observed and recorded but unfortunately not documented at that time.

Lines 388: Does it refer to the R4 that are neither TR4 nor STR4? Please show the results.

This sentence in the discussion part of the MS was modified

Line 390: It is effective against R4 in other geographic areas?

As of now, the results are limited to India, as we have not used other Foc R4 isolates from other countries except 23 Foc reference isolates for validation. Moreover, we don’t have an idea about the configuration of the SIX gene profile and genome size of Foc isolates of other sources.

Line 399: Change 2018 by 2019 in Carvalhais et al. reference.

The citation was modified to Carvalhais et al. [27] as suggested

Line 415: ‘1,288,728 and 1289395nt’. Please, be consistent writing numbers.

Typo corrected

Line 421: Write R1 and R4 instead of race1 and race 4.

Corrected accordingly

References

Check the use of italics for Latin names.

Line 489: remove semicolon and complete the information.

Line 492: Please, add doi (doi: 10.1079/PAVSNNR202015004).

Line 494: remove semicolon and complete the information (FAO food Outlook).

Line 496: Please, add doi (DOI: 10.17660/ActaHortic.2011.897.44).

Line 566: remove capital letter for f. Sp and Cubense (f. sp. cubense is the correct form).

Line 575: remove capital letter for f. Sp and Cubense.

All the above corrections related references have been done

Round 2

Reviewer 3 Report

From the response to the reviewer`s comments, the conclusion can be given that the manuscript meets the requirements for accept.

Author Response

The reviewer suggested that the MS meet the requirements to accept for publication. However we have once gain gone through the entire manuscript and corrected  all the minor mistakes 

Reviewer 4 Report

Authors have accepted all the minor changes proposed by the reviewer, however they have declined to make some changes that required more effort and which in my opinion would improve the paper.
I suggested that primers should be tested with strains from more geographic areas. I am aware that now they cannot performe these experiments. Nevertheless, other issues such as adding information to the introduction, eliminating data from which no results are shown or modifying figure 3 because the legend is not easy to follow by readers, etc., I think they are simple to do and would increase the quality from the paper.

Author Response

As suggested by the reviewer, all the corrections have been carried out
